# Receiver operating characteristic curve analysis of clinical signs for screening of convergence insufficiency in young adults

**Byeong-Yeon Moon, Sang-Yeob Kim, Dong-Sik Yu** *

Department of Optometry, Kangwon National University, Samcheok, Korea

* yds@kangwon.ac.kr

**Data Availability Statement:** All relevant data are within the manuscript and its Supporting Information files.

## Abstract

Convergence insufficiency (CI) is a dysfunction of binocular vision that is associated with various signs and symptoms in near work. However, CI screening is performed less frequently in adults than in children. We aimed to evaluate the ability of screening tests to discriminate CI from other binocular vision anomalies and normal binocular vision in young adults. One hundred eighty-four university students (age, 18–28 years) who underwent an eye examination due to ocular discomfort were included. Near point of convergence (NPC), phoria, accommodative amplitude, fusional vergence, the ratio of accommodative convergence to accommodation, relative accommodation, binocular accommodative facility, vergence facility, and the values corresponding to Sheard's and Percival's criteria were evaluated. Receiver operating characteristic (ROC) curve analysis for each test was also performed. The prevalence of CI ranged from 10.3% to 21.2%, depending on the signs and the presence of CI associated with accommodative disorders. Assessments based on NPC, Sheard's criterion, and Percival's criterion showed high discriminative ability, with the ability being higher between the CI and normal binocular vision groups than between the CI and non-CI groups. Sheard's criterion showed the highest diagnostic performance in discriminating CI with three signs from the non-CI group. The cut-off values were 7.2 cm for NPC, -0.23 to 1.00 for Sheard's criterion, and -4.00 to -2.33 for Percival's criterion. Our results suggest that the use of Sheard's criterion with NPC shows high performance for screening of CI.

## Introduction

Non-strabismic binocular vision anomalies consist mostly of accommodative and vergence disorders. Vergence disorders are classified into convergence insufficiency (CI), convergence excess, divergence insufficiency, divergence excess, basic esophoria, basic exophoria, and fusional vergence dysfunctions based on phoria and the accommodative convergence to accommodation ratio (AC/A) [1, 2]. CI is a dysfunction of binocular vision that is associated with various symptoms, including blurred vision, diplopia, and discomfort while performing near work [3]. The signs of CI include exophoria that is greater at near than at distance, reduced near point of convergence (NPC), reduced positive fusional vergence (PFV), low

**Funding:** The authors received no specific funding for this work.

**Competing interests:** The authors have declared that no competing interests exist.

AC/A, and deficiencies in negative relative accommodation (NRA) [1]. However, cases with CI can be simple or complex aspects because not all patients with CI have all these symptoms and signs [1, 4]. The reported prevalence of CI varies widely from 1.0% to 33% [1, 5–7]. These wide variations can be attributed to the differences in patient age, sample populations, and diagnostic criteria used in the previous studies [8]. Therefore, it is important to evaluate the ability of clinical diagnostic tests to discriminate between CI and other disorders, since the clinical criteria used for screening CI have been different across several studies.

Receiver operating characteristic (ROC) curve analysis is useful for evaluating the quality or performance of diagnostic tests. In a very recent study [9], ROC curve analysis was used to assess the performance of PFV, NPC, AC/A, accommodative amplitude (AA), binocular accommodative facility (BAF), ratio of PFV over the phoria (PFV/phoria), and the convergence insufficiency symptom survey (CISS) to discriminate between school-age children with and without CI. The results of that study indicated that the NPC break performed best in identifying children with CI. However, these results cannot be extended to adults since that study was limited to patients aged 9 to 18 years and because of the limited validity of the Convergence Insufficiency Symptom Survey (CISS) used for children [10]. A previous study used ROC analysis to investigate the diagnostic validity of the clinical signs of CI in participants aged 19 to 35 years with either symptomatic large exophoria and normal heterophoria, or low visual discomfort at near [11]. In the study, NPC and BAF tests showed the best diagnostic accuracy for discriminating between the two groups. In another ROC curve analysis of only BAF and vergence facility (VF) in university students with symptomatic CI and normal participants classified using the CISS [12], BAF showed greater accuracy than VF for diagnosis of CI. All these previous studies have evaluated the validity of diagnostic tests between symptomatic patients with CI and normal participants. However, in addition to cases of independent CI, CI can present with signs associated with other vergence disorders or accommodative disorders in clinical diagnoses of non-strabismic binocular vision anomalies [13]. The symptoms of CI are common in adults, but they often do not appear [1, 5]. Thus, a careful differential diagnosis is required to identify this condition.

The first consideration for screening CI in the presence of non-strabismic binocular vision anomalies is to distinguish it from accommodative dysfunction through analysis of the values or signs measured by each test for accommodative and vergence functions used in the evaluation of binocular vision. The second approach involves identification of CI-related signs. The final approach requires identification of a close correlation between symptoms and clinical signs. In the final approach, however, the symptoms may not be related to clinical signs, especially in adults. Many symptomatic CI disorders are related to the presence of defects in two or more areas of binocular vision function [13, 14].

Our study was limited to young adults and an approach focused on signs and modified signs such as Sheard's and Percival criterion, in addition to fusional vergence related to ability to maintain single binocular vision. The purpose of this study was to evaluate the accommodative and vergence ability for university students who visited primary eye care due to ocular discomfort and to analyze the diagnostic ability of each test for CI screening by performing ROC curve analysis, including assessments of sensitivity, specificity, cut-off values, and likelihood ratio.

## Materials and methods

### Subjects

The participants were 184 university students (age = 18–28 years; mean age = 22.23 ± 2.26 years) who underwent an eye examination due to ocular discomfort. Participating students

voluntarily visited a university eye clinic center for primary eye care due to blurred vision, eye-strain, and visual discomfort. This study was approved by the Kangwon National University Institutional Review Board (KWNUIRB-2019-02-001-002), with waiver of the informed consent for the retrospective collection of clinical data, and adhered to the tenets of the Declaration of Helsinki. Participants had not previously undergone any vison therapy or eye exercise treatment. We excluded patients with ocular diseases such as glaucoma, cataract, and retinal disease, and those with a history of prior surgery, which was determined by history-taking [1]. The criteria for inclusion into study also were the absence of amblyopia and strabismus.

## General procedure

Before refraction assessments, all participants underwent a case evaluation to obtain information about the ocular discomfort, followed by ocular motility testing and a cover-uncover test to rule out strabismus. Refraction and binocular vision tests were performed using a phoropter (VT-SE; Topcon Corporation, Tokyo, Japan) and visual charts (ACP-8; Topcon Corporation, Tokyo, Japan) at distance (6 m) and near (40 cm). All measurements were taken in a general clinical room by the same examiner, who performed all tests within approximately 30 minutes, using the same methodology.

## Assessment of binocular vison function

Following pre-refraction and the best-corrected refraction of ≥20/20 visual acuity, the NPC was measured in free space by using an isolated 20/50 target on a Gulden fixation stick to evaluate the subjects' ability to converge the eyes while retaining binocular single vision [15]. The measured value was determined by the distance (cm) from the subject's eye or spectacle plane at which the subject reported double vision when the target was moved slowly toward the subject between both eyes and at eye level. The normative value for the break point of NPC is 5 ± 2.5 cm.

Phoria was measured by using the von Graefe technique [16] to assess the direction and amplitude of eye alignment, which indicated latent misalignment of the eyes. Horizontal phoria was measured with a measuring prism of 12 Δ (prism) base in (BI) and a dissociating prism of 6 Δ base up (BU) before one eye and the fellow eye, respectively. The measured value was the amount and direction of the prism when the measuring prism of 12 Δ BI was reduced slowly until the subject reported vertical alignment of the lower and upper 20/30 letter target separated by the two prisms. After measurement of phoria at distance, near phoria was measured. Negative values represent exophoria, whereas positive values represent esophoria. The normative values for distance phoria and near phoria are 1exo ± 1 Δ and 3exo ± 3 Δ, respectively.

Binocular AA is the maximum amount of accommodation under binocular conditions, which represents the visual function required for maintaining a clear image [17]. AA was measured by the push-up method by using an accommodative convergence rule (GR50, Bernell, USA) and a near target (near visual acuity, 20/30). The near point of accommodation was determined by assessing the distance from the subject's eye or spectacle plane to the target at which the subject reported when the target first became blurred while moving at a rate of approximately 2 cm/s towards the subject's eyes. The amount of AA was expressed in diopter (D) by the inverse of the near point of accommodation (m). The normative value for mean AA is 18.5 –(1/3) age.

Fusional vergences were measured by using the rotary prism on the phoropter and a 20/30 letter target to evaluate the subject's ability to use horizontal vergence to maintain binocular vision. First, negative fusional vergences (NFVs) at distance were measured by using the BI

prism [17]. The prism power was increased in front of both eyes at the rate of 1–2 Δ/sec until subjects reported a blurred image (blur point) and then a double image (break point), and then decreased to a find a recovery point until the subject reported a single image. PFVs at distance were measured by base out (BO) prism with the same method as NFVs. Fusional vergences at near were measured after fusional vergence tests at distance [17]. If no blur value in fusional vergence was reported, the break value was used in the analysis. The normative values are 9 ± 2 Δ for bur, 19 ± 4 Δ for break, and 10 ± 2 Δ for recovery of PFV at distance; 7 ± 2 Δ for break, and 4 ± 1 Δ for recovery of NFV at distance; 17 ± 3 Δ for bur, 21 ± 3 Δ for break, and 11 ± 4 Δ for recovery of PFV at near; 13 ± 2 Δ for bur, 21 ± 2 Δ for break, and 13 ± 3 Δ for recovery of NFV at near.

Gradient AC/A ratio (AC/A) was determined as the difference in phoria between before and after the addition of + 1.00 D at near divided by 1.00 D. Calculated AC/A was determined as the sum of inter-pupillary distance (cm), measured by PD meter, and the difference in phoria between near and distance divided by 2.50 D [17, 18]. The normative value for AC/A is 4 ± 1 Δ.

Relative accommodations were measured to examine the subject's ability to increase and decrease accommodation under binocular vision when the convergence demand was constant [19]. Negative relative accommodation (NRA) was measured first, adding plus power over the refraction at the rate of 0.25 D/2 s. The measured values were the amount of plus power added until the subject reported the first maintained blur. Positive relative accommodation (PRA) was measured as the minus power added over the refraction until the sustained blur. The normative values for NRA and PRA are + 2.00 ± 0.25 D and -2.37 ± 0.62 D, respectively.

BAF was measured at 40 cm by using a ± 2.00 D binocular flipper lens to evaluate the ability of the accommodative response at near [20]. If the subject reported clear vision for a 20/30 letter target when +2.00 D was placed in both eyes, the lens was flipped to place -2.00 D in both eyes. When the subject reported clear vision again, this indicated one cycle. Measured values were in terms of the number of cycles per minute (cpm). The normative value for BAF is > 13 cpm.

Vergence facility (VF) was measured by using a prism flipper (3 Δ BI + 12 Δ BO) to evaluate the ability of the fusional vergence at near (40 cm) [21]. If the subject reported single vision with a 20/30 letter target when 12 Δ BO was placed in both eyes, the prism was flipped to place the 3 Δ BI in both eyes. When the subject reported clear vision again, this indicated one cycle. Measured values were in terms of the number of cycles per minute (cpm). The normative value for VF is > 12 cpm.

The relation of fusional vergence to phoria was analyzed by using Sheard's [22] and Perival's criteria [23]. Sheard's criterion aimed to determine whether the blur point of fusional vergence is at least twice the phoria. The values for Sheard's criterion were calculated as 2/3 the phoria minus 1/3 the fusional vergence. Vergence anomalies were considered to exist when the subjects failed to meet this criterion (value of Sheard's criterion > 0). Percival's criterion was that the orthophoria point of subject should be operating in the middle-third of the binocular vergence range. The values for Percival's criterion were determined by calculating the value 1/3 greater of two fusional vergences minus 2/3 lesser of the two fusional vergences. If the obtained value was positive, vergence anomalies were considered to be present.

All data were compared with Morgan's expected findings [17, 24], and the main characteristic signs were identified to diagnose accommodative and vergence anomalies. The diagnosis of accommodative and vergence anomalies was classified by signs based on Table 1, which referred to Scheiman and Wick's study [17], and compared with the expected criteria for each test [24]. Subjects groups were classified into three groups according to Table 1: All CI including CI with two signs and CI with an accommodative dysfunction, other binocular vision

**Table 1. Diagnostic criteria for non-strabismic binocular vision anomalies.**

| | |
|---|---|
| **Convergence insufficiency (CI)**<br>Presence of sign 1 and sign 2 or 3<br>1. Exophoria at near $\geq 4\,\Delta$ than at distance<br>2. PFV at near $< 14\,\Delta$ for blur or $<18\,\Delta$ for break<br>3. NPC $\geq 7.5$ cm | **Basic esophoria (BE)**<br>Presence of sign 1 and sign 2 or 3<br>1. Esophoria $< 4\,\Delta$ difference between distance and near<br>2. NFV at distance $< 5\,\Delta$ for break & NFV at near $< 11\,\Delta$ for blur or $<19\,\Delta$ for break<br>3. Gradient or calculated AC/A ratio 3–5/1 |
| **Divergence insufficiency (DI)**<br>Presence of sign 1 and sign 2 or 3<br>1. Esophoria at distance $\geq 4\,\Delta$ than at near<br>2. NFV at distance $< 5\,\Delta$ for break<br>3. Gradient or calculated AC/A ratio $< 3/1$ | **Fusional vergence dysfunctions (FVD)**<br>Presence of sign 1 and sign 2 or 3<br>1. Normal phoria at distance (1 exo $\pm$ 1$\Delta$) and near (3 exo $\pm$ 3$\Delta$)<br>2. Reduced (low) PFV and NFV at distance and near<br>3. Gradient or calculated AC/A ratio 3–5/1 |
| **Convergence excess (CE)**<br>Presence of sign 1 and sign 2 or 3<br>1. Esophoria at near $\geq 4\,\Delta$ than at distance<br>2. NFV at near $< 11\,\Delta$ for blur or $<19\,\Delta$ for break<br>3. Gradient or calculated AC/A ratio $> 5/1$ | **Accommodative insufficiency (AI)**<br>Presence of sign 1 and sign 2 or 3<br>1. Reduced binocular AA for minimum amplitude = $15.0 - 0.25 \times$ age<br>2. Low PRA $< 1.75$ D<br>3. BAF $< 13$ cpm |
| **Divergence excess (DE)**<br>Presence of sign 1 and sign 2 or 3<br>1. Exophoria at distance $\geq 4\,\Delta$ than at near<br>2. PFV at distance $< 7\,\Delta$ for blur or $<15\,\Delta$ for break<br>3. Gradient or calculated AC/A ratio $> 5/1$ | **Accommodative excess (AE)**<br>Presence of signs 1 and 2<br>1. Low NRA $< 1.75$ D<br>2. BAF $< 13$ cpm |
| **Basic exophoria (BX)**<br>Presence of sign 1 and sign 2 or 3<br>1. Exophoria $< 4\,\Delta$ difference between distance and near<br>2. PFV at distance $< 7\,\Delta$ for blur or $<15\,\Delta$ for break, and PFV at near $< 14\,\Delta$ for blur or $<18\,\Delta$ for break<br>3. Gradient or calculated AC/A ratio 3–5/1 | **Accommodative infacility**<br>Presence of signs 1 and 2<br>1. Low NRA & PRA $< 1.75$ D<br>2. BAF $< 13$ cpm |

D: diopter, $\Delta$: prism diopter, cpm: cycles per minute, AA: accommodative amplitude, NPC: near point of convergence, AC/A: accommodative convergence/accommodation, NFV: negative fusional vergence, PFV: positive fusional vergence, NRA: negative relative accommodation, PRA: positive relative accommodation, BAF: binocular accommodative facility.

anomaly (BVA) except CI, and normal binocular vision (NBV). All CI were also classified into CI with three signs (CI3), CI with two signs and an accommodative dysfunction (CI3AD), and CI with two signs (CI2).

## Data analysis

Data were collected and analyzed using IBM SPSS Statistics version 19 (IBM Corp., USA). The descriptive statistical analysis was based on mean and standard deviation values. Prior to ANOVA analysis, the data was checked for normality using the Kolmogorov–Smirnov test. A Kruskal–Wallis one-way ANOVA (nonparametric test) with a Dunn–Bonferroni's *post-hoc* test comparing the three groups (all CI, BVA, and NBV) was performed. A p-value of $< 0.05$ was considered significant. ROC curve analysis was performed by plotting sensitivity on the y axis as a function of 1 –specificity on the x axis to analyze the diagnostic ability of each test (AA, NPC, AC/A, NFV, PFV, NRA, PRA, BAF, VF, Sheard's and Percival's criterion) in screening CI. Sensitivity refers to the probability that a test will indicate CI when CI is present, and specificity refers to the probability that a test will indicate the absence of CI when CI is not present. The area under the curve (AUC) in the ROC curve analysis indicates the discriminative ability to distinguish between subjects with and without CI. The cut-off value for each test

was defined as the coordinate that had the maximized sum of sensitivity and specificity. The cut-off was also identified for each test with the largest AUC which was significantly greater than 0.50. The likelihood ratio indicates the degree of increase or decrease in the probability of the CI if the test yields positive or negative findings.

## Results

In an assessment of the refractive errors in the participants, of the 368 eyes, 258 (70.1%) showed myopia (spherical equivalent [SE] = -3.31 ± 2.16 D), 19 (5.2%) showed hyperopia (SE = +0.80 ± 0.51 D), and 91 (24.7%) showed emmetropia. Myopia was present in 125 participants (67.9%), hyperopia in six participants (3.3%), emmetropia in 42 participants (22.8%), mixed anisometropia in four participants (2.2%), simple myopic anisometropia in four participants (2.2%), and simple hyperopic anisometropia in three participants (1.6%).

The means and standard deviations for each test of binocular vision function after wearing refractive correction are shown in Table 2. Subjects were divided into three groups: all CI (n = 39); binocular vision anomalies (BVAs; n = 49) except CI; and normal binocular vision (NBV; n = 96). The Kruskal–Wallis one-way ANOVA with Dunn–Bonferroni's *post-hoc* test showed significant intergroup differences in NPC, near phoria, and calculated AC/A; significant differences between all CI and other BVA, and other BVA and NBV in distance phoria and distance NFV; and significant differences between all CI and NBV in gradient AC/A, PFV blur and recovery at distance, near PFV, NRA, PRA, BAF, and VF.

Table 3 shows the prevalence of binocular vision anomalies diagnosed according to Table 1 based on data for each test prior to the ROC curve analysis. It also shows the characteristics of the subjects enrolled in CI screening assessments. Of these 184 young adults, 96 (52.2%) were classified as NBV and 34 (18.5%) were identified as having CI2. CI with accommodative disorders (CI + AI and CI + AE) was observed in five (2.7%) participants, while the prevalence of CI varied from 10.3% (19 participants with CI3) to 21.2% (39 participants with CI2AD) depending on the number of signs and the presence of associated accommodative disorders. When participants with BVAs other than CI were referred to as non-CI, the distribution between CI and non-CI was different according to the classification criteria.

ROC curve analysis was performed for the tests shown in Table 2. The results of the AUC for NPC, Sheard's and Percival's criterion with p < 0.05 and AUC > 0.5 in 95% confidence interval ranges, and PFV, NFV, and AC/A including the diagnostic criteria in Table 1 are shown in Table 4. Fig 1 shows ROC curves for NPC, Sheard's and Percival's criterion with statistically significant differences when comparing the AUC to the value 0.5. Among the AUCs for all diagnostic tests included in Table 1, the values were significantly greater than 0.5 for only the NPC and NFV at distance, and the AUCs for diagnostic tests excluded in Table 1 were also significantly greater than 0.5 for Sheard's and Percival's criteria. AUCs were greater for NBV than for non-CI, greater for excluding than for including CI associated with AD, and greater for CI with three signs than for CI with two signs.

Table 5 shows the sensitivity, specificity, positive likelihood ratio (LR+), and negative likelihood ratio (LR-) for each test by using cut-offs obtained with ROC curves. The NPC cut-offs were >7.2 cm for all classified CIs combined with the NBV and non-CI groups. The Sheard's criterion cut-off of >1.00 for CI with three signs (CI3) was higher than the cut-off of >-0.23 for CI with two signs (CI2), and showed higher sensitivity and specificity than CI2. The Percival criterion cut-off of >-4.00 for CI2 was lower than the cut-off of >-2.33 for CI3. Sensitivity, specificity, and LR+ were higher in the order of NPC, Sheard's, and Percival's criterion, but LR- showed the opposite trend.

**Table 2. Comparison of mean and standard deviation values for the measures of binocular function.**

| Test | All subjects (n = 184) | Subject groups | | | |
|---|---|---|---|---|---|
| | | a. All CI (n = 39) | b. Other BVA (n = 49) | c. NBV (n = 96) | p-value (*post-hoc*) |
| AA (D) | 11.46 ± 2.48 | 10.90 ± 1.75 | 11.24 ± 2.83 | 11.79 ± 2.51 | 0.198 |
| NPC (cm) | 7.29 ± 1.93 | 9.21 ±2.07 | 7.24 ± 1.85 | 6.53 ± 1.27 | < 0.001* (a>b>c) |
| Phoria at distance (Δ)† | -2.28 ± 2.91 | -3.04 ± 2.10 | -0.88 ± 4.04 | -2.69 ± 2.20 | < 0.001* (a, c>b) |
| Phoria at near (Δ)† | -6.34 ± 5.63 | -10.37 ± 2.84 | -1.71 ± 5.93 | -7.10 ± 4.63 | < 0.001* (a>c>b) |
| Calculated AC/A (Δ/D) | 4.82 ± 1.82 | 3.55 ± 1.33 | 6.16 ± 2.06 | 4.66 ± 1.38 | < 0.001* (b>c>a) |
| Gradient AC/A (Δ/D) | 3.14 ± 1.62 | 2.40 ± 1.42 | 3.54 ± 1.73 | 3.23 ± 1.55 | 0.010* (b, c>a) |
| NFV break at distance (Δ) | 9.11 ± 3.99 | 9.72 ± 3.80 | 7.78 ± 3.64 | 9.54 ± 4.12 | 0.025* (a, c>b) |
| NFV recovery at distance (Δ) | 5.28 ± 2.91 | 6.15 ± 3.10 | 4.25 ± 2.45 | 5.46 ± 2.92 | 0.011* (a, c>b) |
| PFV blur at distance (Δ) | 10.08 ± 4.51 | 8.69 ± 3.18 | 9.02 ± 3.95 | 11.19 ± 4.97 | 0.004* (c>a, b) |
| PFV break at distance (Δ) | 16.16 ± 7.84 | 14.62 ± 5.95 | 14.04 ± 7.40 | 17.87 ± 8.39 | 0.005* (c>a, b) |
| PFV recovery at distance (Δ) | 9.76 ± 6.57 | 8.03 ± 4.68 | 8.55 ± 5.87 | 11.08 ± 7.30 | 0.023* (c>a, b) |
| NFV blur at near (Δ) | 13.47 ± 5.19 | 14.10 ± 5.00 | 12.10 ± 5.72 | 13.91 ± 4.91 | 0.236 |
| NFV break at near (Δ) | 19.92 ± 6.55 | 21.00 ± 5.67 | 18.25 ± 7.84 | 20.33 ± 6.05 | 0.241 |
| NFV recovery at near (Δ) | 13.51 ± 5.85 | 14.03 ± 4.99 | 12.47 ± 6.80 | 13.82 ± 5.65 | 0.380 |
| PFV blur at near (Δ) | 15.52 ± 6.23 | 13.41 ± 6.41 | 13.41 ± 6.36 | 17.45 ± 5.47 | < 0.001* (c>a, b) |
| PFV break at near (Δ) | 21.15 ± 8.48 | 17.92 ± 8.10 | 19.08 ± 8.63 | 23.51 ± 7.90 | < 0.001* (c>a, b) |
| PFV recovery at near (Δ) | 13.39 ± 8.70 | 11.10 ± 8.71 | 11.43 ± 7.61 | 15.32 ± 8.85 | 0.005* (c>a, b) |
| NRA (D) | 2.15 ± 0.63 | 1.87 ± 0.65 | 1.92 ± 0.65 | 2.39 ± 0.50 | < 0.001* (c>a, b) |
| PRA (D) | -2.77 ± 1.15 | -2.76 ± 1.26 | -2.40 ± 1.11 | -2.98 ± 1.10 | 0.005* (c>b) |
| BAF (cpm) | 12.47 ± 5.64 | 10.74 ± 5.61 | 12.12 ± 6.63 | 13.35 ± 4.95 | 0.024* (c>a) |
| VF (cpm) | 13.27 ± 3.93 | 10.59 ± 4.09 | 13.61 ± 4.04 | 14.19 ± 3.31 | < 0.001* (b, c>a) |
| Sheard's criterion at distance | -1.22 ± 2.00 | -0.87 ± 1.80 | -0.30 ± 2.15 | -1.83 ± 1.80 | < 0.001* (a, b>c) |
| Sheard's criterion at near | -0.15 ± 3.25 | 2.44 ± 2.75 | -0.69 ± 2.98 | -0.92 ± 3.05 | < 0.001* (a>b, c) |
| Percival's criterion at distance | -1.11 ± 1.69 | -1.27 ± 1.39 | -0.54 ± 1.60 | -1.32 ± 1.80 | 0.013* (b>a, c) |
| Percival's criterion at near | -1.77 ± 2.34 | -1.11 ± 2.32 | -1.23 ± 2.34 | -2.32 ± 2.23 | 0.006* (a, b>c) |

CI: convergence insufficiency, BVA: binocular vision anomaly, NBV: normal binocular vision, D: diopter, Δ: prism diopter, cpm: cycles per minute, AA: accommodative amplitude, NPC: near point of convergence, AC/A: accommodative convergence/accommodation, NFV: negative fusional vergence, PFV: positive fusional vergence, NRA: negative relative accommodation, PRA: positive relative accommodation, BAF: binocular accommodative facility, VF: vergence facility.

*p < 0.05 indicates statistically significant differences among groups in Kruskal–Wallis one-way ANOVA followed by Dunn–Bonferroni's *post-hoc* test. †Minus and plus signs in phoria indicate exophoria and esophoria, respectively.

## Discussion

In this study, with ROC curve analysis of signs, the tests that showed a significant discriminative ability between CI and non-CI or CI and NBV among young adult university students were those based on the Sheard's and Percival's criteria for near vision, while NPC assessment was the best diagnostic test for identifying CI. For CI screening between CI3 (CI with three signs) and non-CI, Sheard's criterion was a better diagnostic parameter than NPC. The distribution of CI, non-CI, and NBV according to diagnostic criteria such as population and the signs of CI influenced the validity of each test in the ROC curve analysis.

The prevalence of CI in this study ranged between 10.3% and 21.2%, higher than rates of 1.5% to 10.8% reported in previous studies of the general population–university students [8, 25, 26]. These variations in the prevalence of CI can be attributed to differences in methodological aspects, including instrumentation and techniques, classification criteria, and the number of diagnostic signs; the types of populations studied (clinical/non-clinical); data analysis

**Table 3. Prevalence of binocular vision anomalies.**

| Dysfunction | n | % | CI subjects associated with signs and AD |
|---|---|---|---|
| CI | 34 | 18.5 | Classified CI vs. non-CI group[†] |
| DI | 11 | 6.0 | CI + 3 signs (CI3): 19 (10.3%) vs. 165 (89.7%) |
| CE | 4 | 2.2 | CI + 3 signs + AD (CI3AD): 23 (12.5%) vs. 161 (87.5%) |
| DE | 2 | 1.1 | CI + 2 signs (CI2): 34 (18.5%) vs. 150 (81.5%) |
| BX | 9 | 4.9 | CI + 2 signs + AD (CI2AD): 39 (21.2%) vs. 145 (78.8%) |
| BE | 4 | 2.2 | |
| AI | 7 | 3.8 | |
| AE | 2 | 1.1 | |
| CI + AI | 4 | 2.2 | |
| CI + AE | 1 | 0.5 | |
| CE + AE | 3 | 1.6 | |
| BX + AI | 1 | 0.5 | |
| BE + AI | 1 | 0.5 | |
| Indefinite | 5 | 2.7 | |
| NBV | 96 | 52.2 | |
| Total | 184 | 100 | |

Abbreviations are the same as those in the notes for Table 1. Signs are sign 1, 2, 3 for CI in Table 1. AD: accommodative dysfunction, CI3: CI with 3 signs, CI3AD: CI with 3 signs + CI with an accommodative dysfunction, CI2: CI with 2 signs, CI2AD: CI with 2 signs + CI with an accommodative dysfunction. [†]Non-CI group is a group excluding CI group.

methods; and participant factors including age as well as refractive errors [27]. Our study was performed in university students who underwent primary eye care due to ocular discomfort, and CI was classified based on signs. The CI screening ability of each test was also evaluated in NBV and non-CI conditions. The variation in CI prevalence in this study appears to be associated with differences in signs, classification criteria, and population. In our study, the prevalence of myopia was high (70.1%). CI has been reported to show a significant association with myopia [28], whereas CI and refractive errors were not significantly associated [29]. Although two previous studies showed different results, the high prevalence of CI in this study may be partially explained by the association with myopia. CI may present as types associated with

**Table 4. Results for the area under the curve (AUC) in ROC curve analysis among CI, non-CI, and NBV groups.**

| Ability to discriminate CI | NPC | Sheard[†] | Percival[†] | NFV | PFV[ns] | AC/A[ns] |
|---|---|---|---|---|---|---|
| From CI3 + NBV | 0.920 | 0.912 | 0.717 | (0.627) rec | < 0.2 | < 0.3 |
| From CI3 + non-CI | 0.842 | 0.905 | 0.672 | (0.654) rec | < 0.2 | < 0.3 |
| From CI3AD + NBV | 0.920 | 0.905 | 0.702 | 0.577[ns] brk | < 0.2 | < 0.3 |
| From CI3AD + non-CI | 0.853 | 0.900 | 0.657 | 0.601[ns] brk | < 0.2 | < 0.3 |
| From CI2 + NBV | 0.914 | 0.793 | 0.652 | (0.614) rec | < 0.4 | < 0.3 |
| From CI2 + non-CI | 0.866 | 0.773 | 0.608 | (0.656) rec | < 0.5 | < 0.3 |
| From CI2AD + NBV | 0.913 | 0.804 | 0.642 | (0.566)[ns] rec | < 0.4 | < 0.4 |
| From CI2AD + non-CI | 0.876 | 0.795 | 0.598 | (0.603) rec | < 0.4 | < 0.4 |

Abbreviations are the same as those in the notes for Tables 1, 2 and 3.

[†]Criterion for near, brk: break point, rec: recovery point

[ns]: not significant, (): value for distance.

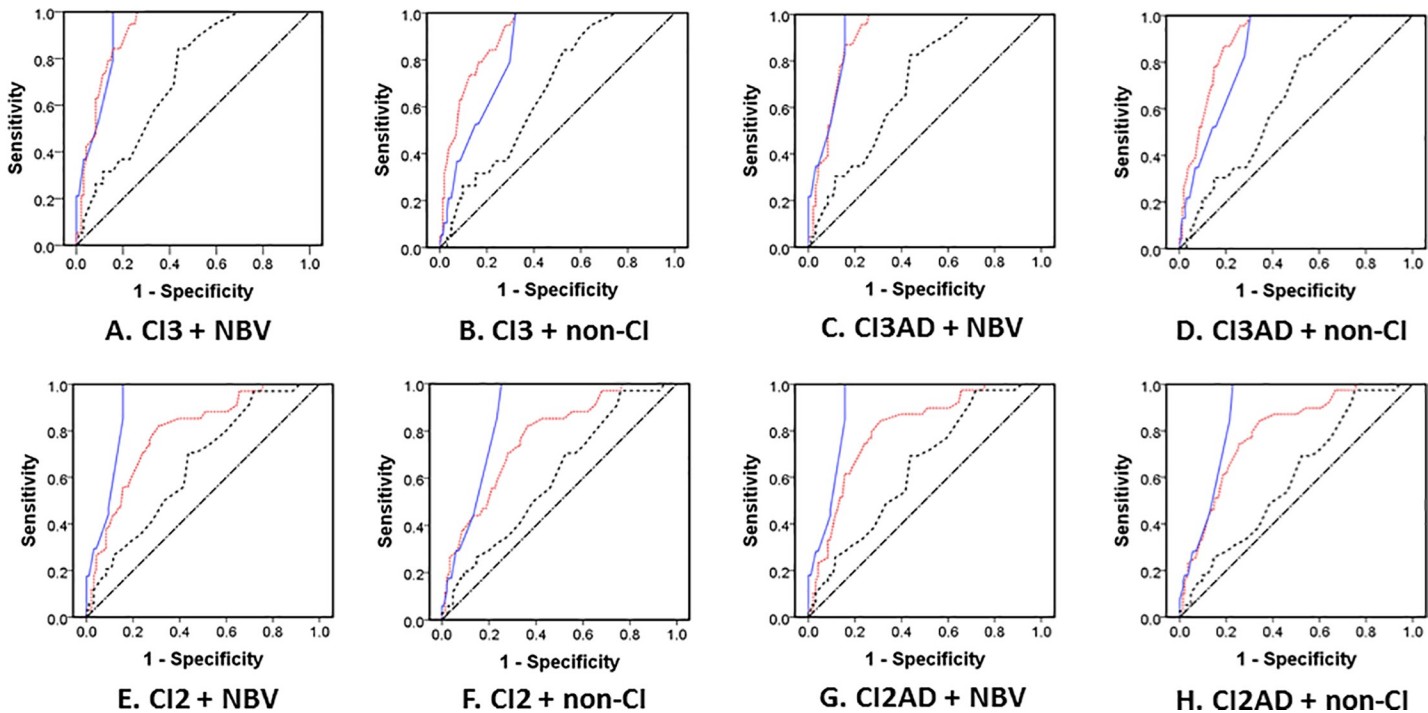

**Fig 1. ROC curves for NPC and Sheard's and Percival's criterion according to the combined CI groups.** Blue solid line: NPC, red dotted line: Sheard's criterion for near, dashed line: Percival's criterion for near, diagonal line: no discrimination (AUC = 0.5).

accommodative dysfunction such as CI plus AI [30] or CI plus accommodative excess (AE) [19]. CI combined with accommodative functions can also exist because the vergence and accommodative systems are linked [31]. CI should also be distinguished from conditions such as normal vision, diverse vergence and accommodative disorders, and other binocular anomalies. Our study factored these conditions to investigate the discriminative ability of each test for CI screening by performing ROC curve analysis under these conditions.

In comparisons of the means and standard deviations for each test, assessments based on NPC, AC/A, PFV blur/break/recovery at near, NRA, BAF, and VF showed significant differences between all-CI and NBV groups, and those based on NPC and AC/A showed significant differences between all-CI and other BVA groups. Tests with a significant difference were noted less frequently in comparisons with other BVA than in comparisons with NBV, as shown in Table 2. In Cacho-Martínez et al.'s [11] study comparing large phoria (>6 Δ) and normal phoria, significant differences were found on 5 tests such as NPC, PFV blur at near, NRA, BAF, and VF. However, in our study comparing all-CI and other BVA groups, significant differences were found on 2 tests of NPC and AC/A. This finding indicates that it is difficult to distinguish CI from other BVA. Since patients with CI must be distinguished from subjects with various other clinical conditions, the efficiency of CI screening depends on the performance of the diagnostic test in discriminating CI from abnormal groups combined with other binocular anomalies rather than normal groups. Tests for the diagnosis of CI were based on the ratio of positive fusional vergence to phoria [9] and Sheard's criterion in other studies [22, 32], and these tests was limited to school-age children. However, Percival's criterion as well as Sheard's criterion was applied to adults in this study. There are significant differences between the all-CI and NBV groups for Sheard's criterion at distance/near and Percival's criterion at near and between the all-CI and other-BVA groups for Sheard's criterion at near.

**Table 5. Diagnostic validity of NPC and Sheard's and Percival's criterion by using cut-offs derived from ROC curve analysis.**

| Screening for combined CI groups | | | Cut-off | Sensitivity | Specificity | LR+ | LR- |
|---|---|---|---|---|---|---|---|
| CI3 | + NBV | NPC | >7.20 | 1.00 | 0.84 | 6.40 | 0.00 |
| | | Sheard | >0.33 | 1.00 | 0.74 | 3.84 | 0.00 |
| | | Percival | >-2.33 | 0.84 | 0.56 | 1.92 | 0.28 |
| | + non-CI | NPC | >7.20 | 1.00 | 0.68 | 3.11 | 0.00 |
| | | Sheard | >0.33 | 1.00 | 0.67 | 3.06 | 0.00 |
| | | Percival | >-2.33 | 0.84 | 0.47 | 1.60 | 0.33 |
| CI3AD | + NBV | NPC | >7.20 | 1.00 | 0.84 | 6.40 | 0.00 |
| | | Sheard | >0.33 | 1.00 | 0.74 | 3.84 | 0.00 |
| | | Percival | >-2.33 | 0.83 | 0.56 | 1.89 | 0.31 |
| | + non-CI | NPC | >7.20 | 1.00 | 0.70 | 3.29 | 0.00 |
| | | Sheard | >1.00 | 0.96 | 0.74 | 3.67 | 0.06 |
| | | Percival | >-2.33 | 0.83 | 0.48 | 1.58 | 0.36 |
| CI2 | + NBV | NPC | >7.20 | 1.00 | 0.84 | 6.40 | 0.00 |
| | | Sheard | >-0.23 | 0.82 | 0.69 | 2.64 | 0.26 |
| | | Percival | >-2.33 | 0.71 | 0.56 | 1.61 | 0.52 |
| | + non-CI | NPC | >7.20 | 1.00 | 0.75 | 3.95 | 0.00 |
| | | Sheard | >-0.23 | 0.82 | 0.63 | 2.25 | 0.28 |
| | | Percival | >-4.00 | 0.97 | 0.24 | 1.28 | 0.12 |
| CI2AD | + NBV | NPC | >7.20 | 1.00 | 0.84 | 6.40 | 0.00 |
| | | Sheard | >-0.23 | 0.85 | 0.69 | 2.71 | 0.22 |
| | | Percival | >-4.00 | 0.97 | 0.28 | 1.36 | 0.09 |
| | + non-CI | NPC | >7.20 | 1.00 | 0.77 | 4.39 | 0.00 |
| | | Sheard | >-0.23 | 0.85 | 0.66 | 2.45 | 0.23 |
| | | Percival | >-4.00 | 0.97 | 0.25 | 1.30 | 0.10 |

Abbreviations are the same as those in the notes for Tables 1, 2 and 3. LR+: positive likelihood ratio, LR-: negative likelihood ratio.

These results indicate that Sheard's criterion could be used in tests to distinguish CI from other BVA and NBV, and Percival's criterion could be used to distinguish CI from NBV. However, Sheard's criterion is a useful tool for screening CI with exophoria associated with near tasks because the signs of CI include exophoria more than 6 Δ at near and normal phoria of 0–6 Δ exophoria at distance [17], and previous studies [33, 34] have suggested that Sheard's and Percival's criteria are the most effective with exophoria and esophoria, respectively.

The main finding in this study was that NPC can distinguish individuals with CI signs or CI signs associated with AD, namely, CI2AD and CI3AD from the NBV and non-CI groups. In ROC curve analysis, the AUC of 0.842–0.920 obtained using the NPC test represents an excellent discriminative ability for CI screening. Although the test parameters such as subjects and diagnostic and classification criteria were not consistent with other studies [25, 26], our result is consistent with outcomes that suggest NPC is the best in identifying children with CI [9] and subjects with large near exophoria and moderate to severe symptoms [11]. The discriminative ability in CI groups (CI3, CI3AD, CI2, CI2AD) combined with NBV is higher than that in CI groups combined with non-CI. The NPC test showed a cut-off value of >7.2 cm in comparison with previously reported values of 7 cm [11] or 7.5 cm [35, 36] and sensitivity of 1.00 in all groups, and specificity and positive likelihood ratio (LR+) were higher in the combined group of CI and NBV than in the combined group of CI and non-CI. A high LR+ indicates a high ratio of the probability of the true presence of CI to the probability of false presence of CI in the NPC test. A low LR- indicates a low ratio of the probability of false

absence of CI to the probability of true absence of CI in the NPC test. A negative likelihood ratio (LR-) of zero in all classified CIs indicates a decreased probability that the NPC test is negative.

The ROC curve analysis in this study showed that Sheard's and Percival's criteria have potential for use as tools for CI screening. Sheard's criterion is particularly useful for CI screening from non-CI than from NBV. In addition, Sheard's criterion can be a better tool than NPC in cases of CI with three signs (CI3, CI3AD). The previous studies [9, 11] did not provide cut-off values, sensitivity, specificity, AUC, and other data from ROC curve analysis for Sheard's and Percival's criterion to screen CI. The AUC of 0.773–0.912 obtained using Sheard's criterion in our study represents an acceptable discriminative ability for CI screening, and the AUC reduced with decreasing signs. Sheard's criterion in this study has positive cut-offs values (failed to normal binocular vision or needed prism) in the CI3 and cut-offs of > -0.23 (approximately cut-offs > zero) in the CI2. Sheard's criterion could diagnose all-CI. The sensitivity of the combined CI3 is higher than that of the combined CI2, and the specificity is lower than sensitivity. Although the LR+ of 2.25–3.84 was lower than the corresponding value for the NPC assessment as a positive test result indicating the presence of CI, and the LR- of 0.22–0.28 was higher than the corresponding value for the NPC assessment as a negative result test indicating the absence of CI, from another perspective, this criterion is a valid tool for discriminating CI with three signs (CI3, CI3AD) from non-CI because the AUC of Sheard's criterion was greater than that in NPC and the LR- of zero in CI3 and CI3AD was equal to that for NPC.

Percival's criterion showed cut-off values of <0 (negative value; meet criterion or not needed prism) in all groups. Although Percival's criterion showed cut-off values for screening CI, the ROC curve analysis values indicate that it is a valid tool for CI screening when considering AUC ≥ 0.598, sensitivity ≥ 0.71, specificity of 0.24–0.56, LR+ ≥ 1.28, and LR- ≤ 0.36. Values of for CI screening showed a mismatch between Percival's criterion (the amount of prism required or a positive cut-off value) and the results of ROC curve analysis (a negative cut-off value) but an approximate match between Sheard's criterion (the amount of prism required or a positive value) and the results of ROC curve analysis (cut-off value of -0.23 close to a positive). These differences could have occurred because Sheard's criterion works best for exophoric conditions such as CI and Percival's criterion tends to work best for near esophoric conditions such as convergence excess [23, 37]. However, Percival's criterion showed a lower discriminative ability than Sheard's criterion for CI screening. In ROC curve analysis, Percival's criterion showed lower AUC and LR+ than Sheard's criterion. Percival's criterion also showed lower sensitivity than Sheard's criterion, except for CI2 combined with non-CI (CI2 + non-CI) and CI2 combined with AD (CI2AD + NBV, CI2AD + non-CI). The LR- of Percival's criterion was higher than that of Sheard's criterion except for CI2 combined with non-CI and combined CI2AD. Although Percival's criterion could be a useful tool for CI screening, this criterion needs to be modified to show better diagnostic accuracy for binocular vision disorders. The ROC curve analysis for each test except the above showed no statistically significant differences for CI screening, but significant differences were noted for BAF in other studies [9, 12]. On the other hand, a high PRA was shown to be the most sensitive sign for CI combined with accommodative excess [19], and Gall at al. [21] found that VF can differentiate symptomatic from asymptomatic patients (not assessed in this study). However, BAF, PRA, and the VF test were not shown to be useful in discriminating CI from the normal or non-CI groups in the results of our study. Thus, factors related to the classification criteria for binocular vision disorders and subject characteristics such as age and population might lead to a different result.

Although this study was conducted with participants reporting ocular discomfort, one limitation of this study is the lack of evaluation about subjective symptoms since the analysis was

based on objective clinical signs. In the previous studies, some adults with CI signs were asymptomatic [38] and there was no further association between the severity of the clinical signs and symptoms in children aged 9 to 17 years [4]. On the other hand, other studies have shown an association between signs and symptoms [9, 39]. These results cannot extend to young adults. Therefore, the relationship between signs and symptoms in university students needs further study.

In summary, this study shows that Sheard's and Percival's criteria are useful tools to discriminate CI in young adults although the NPC test has diagnostic validity for screening subjects with CI signs from not only NBV but also non-CI with AD and other binocular vision disorders. On the other hand, with respect to the AUCs of the ROC curve analysis for CI screening in cases of CI with three signs, Sheard's criteria are significantly greater than NPC and Percival's criteria. In addition, this study suggests that Percival's criterion such as "orthophoria point should be in the middle third of the vergence range" needs to be revised due to showing acceptable sensitivity and specificity for CI screening at a negative cut-off value within the range to meet this criterion.

## Supporting information

**S1 File. All relevant raw data.**
(XLSX)

## Author Contributions

**Conceptualization:** Byeong-Yeon Moon, Dong-Sik Yu.

**Data curation:** Byeong-Yeon Moon, Sang-Yeob Kim, Dong-Sik Yu.

**Formal analysis:** Byeong-Yeon Moon, Sang-Yeob Kim, Dong-Sik Yu.

**Investigation:** Byeong-Yeon Moon, Dong-Sik Yu.

**Methodology:** Byeong-Yeon Moon, Sang-Yeob Kim, Dong-Sik Yu.

**Project administration:** Dong-Sik Yu.

**Resources:** Byeong-Yeon Moon, Dong-Sik Yu.

**Supervision:** Dong-Sik Yu.

**Validation:** Byeong-Yeon Moon, Dong-Sik Yu.

**Visualization:** Byeong-Yeon Moon, Dong-Sik Yu.

**Writing – original draft:** Byeong-Yeon Moon, Sang-Yeob Kim, Dong-Sik Yu.

**Writing – review & editing:** Byeong-Yeon Moon, Sang-Yeob Kim, Dong-Sik Yu.

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
