## [Decision Letter · Decision Letter 0]

9 Sep 2019

PONE-D-19-17984

Receiver operating characteristic curve analysis of clinical signs for screening of convergence insufficiency in young adults

PLOS ONE

Dear Mr. Yu,

Thank you for submitting your manuscript to PLOS ONE. After careful consideration, we feel that it has merit but does not fully meet PLOS ONE’s publication criteria as it currently stands. Therefore, we invite you to submit a revised version of the manuscript that addresses the points raised during the review process.

We would appreciate receiving your revised manuscript by Oct 24 2019 11:59PM. To enhance the reproducibility of your results, we recommend that if applicable you deposit your laboratory protocols in protocols.io, where a protocol can be assigned its own identifier (DOI) such that it can be cited independently in the future. For instructions see: http://journals.plos.org/plosone/s/submission-guidelines#loc-laboratory-protocols

We look forward to receiving your revised manuscript.

Kind regards,

Le Hoang Son, Ph.D

Academic Editor

PLOS ONE

Journal Requirements:

2.  We noticed you have some minor occurrence of overlapping text with the following previous publication, which needs to be addressed: INSUFFI, CONVERGENCE. "Randomized clinical trial of treatments for symptomatic convergence insufficiency in children." Arch Ophthalmol 126.10 (2008): 1336-1349. In your revision ensure you cite all your sources (including your own works), and quote or rephrase any duplicated text outside the methods section. Further consideration is dependent on these concerns being addressed.

3. Please provide additional details regarding participant consent. In the ethics statement in the Methods and online submission information, please ensure that you have specified (1) whether consent was informed and (2) what type you obtained (for instance, written or verbal, and if verbal, how it was documented and witnessed). If the need for consent was waived by the ethics committee, please include this information.

**Comments to the Author**

1. Is the manuscript technically sound, and do the data support the conclusions?

Reviewer #1: Partly

Reviewer #2: Partly

Reviewer #3: Yes

2. Has the statistical analysis been performed appropriately and rigorously? 

Reviewer #1: Yes

Reviewer #2: No

Reviewer #3: N/A

3. Have the authors made all data underlying the findings in their manuscript fully available?

Reviewer #1: Yes

Reviewer #2: Yes

Reviewer #3: Yes

4. Is the manuscript presented in an intelligible fashion and written in standard English?

Reviewer #1: Yes

Reviewer #2: No

Reviewer #3: No

5. Review Comments to the Author

Reviewer #1:

• SUMMARY:

The article provides an analysis of differentiating which clinical tests that are best to be performed to diagnose a common binocular vision anomaly, i.e. the convergence insufficiency (CI). The complexity of the anomaly makes practitioners confused and sometimes missed out in detecting and diagnosing CI. As such, this article highlights the important clinical tests and criteria that could help practitioner by performing the Receiver operating characteristic (ROC) curve analysis. While the ROC analysis were well performed, there are a few issues that needs to be addressed as stated below.

• FEEDBACK:

1. Materials & Methods Section

• Line 99

• Please also clarify if participants recruited have not had any vision therapy or eye exercise treatment prior to the enrollment as research participant.

• Line 112

• The authors should detailed the NPC measurement whether it was done in a free space using accommodative fixation target or using the RAF ruler.

• MAJOR: Target used for the test (20/50) was too large for the 20/20 research participants group. Using this target will not exert the best binocular vergence ability of the participants and would affect the whole test outcomes. Please clarify.

• The authors should specify if the procedures were performed by a sole researcher or a group of researchers. Please also state if the procedures were done in the same order for each participant as some binocular vision assessments are invasive, i.e. will likely introduce fatigue to participants if it is not controlled. This too would affect the whole test outcomes and data collected.

2. Results Section

• Line 207

• Instead of writing the refractive errors and classify them for ‘each eye’, I would suggest that the authors change the way of writing this as per participant, as convergence insufficiency is a binocular anomaly and not a monocular phenomenon.

• Line 219

• It would be beneficial if the authors add in the normative data (e.g.: Morgan’s Normative Data on Binocular Vision) in Table 2 for the readers to understand the distribution of the data against the normative data used clinically.

• Line 269

• The NPC cut-off written in the paragraph was >7.3cm, different than what is stated in Table 5 (>7.2cm). Please double-check.

• The sensitivity and specificity for ROC analysis are usually expressed in percentage with 95% confidence interval. The authors may want to revise how they written this to make it more readable for other researchers.

3. Discussion Section

• Although the participants’ groupings and parameters are complicated but the authors managed to write and explain in a good flow. I hope to see a more analytical explanation when discussing the outputs and its relationship to what have been the current knowledge in diagnosing convergence insufficiency. Some comments I have stated above may or may not change the outputs and interpretation of the results.

4. Grammatical errors (see attachment)

Reviewer #2:

Generally, the Introduction is well-written. Within the Methods there are missing details which mean that the study could not be reproduced. The Results is the weakest section. It is not clear which participants are included in which groups and the names of the groups were changed throughout the results. Please see the comments below for details. It was difficult to follow the discussion due to the concerns regarding the results. I would recommend making those changes first to allow the reviewer to make more informed comments regarding the discussion.

Introduction

44 – delete ‘common’. There is no evidence provided and later on the prevalence of CI is covered with references.

46-8 – no references for the signs of CI

65-67 – study 11 not clearly presented. Did their participants have CI? Were they symptomatic? Was ROC curve analysis used? It is not clear how this study is relevant.

69 – did study 12 test NPC? This was the test that showed best diagnostic ability in the other studies. I would assume this would be tested in this study investigating CI.

77-84 This is an unclear p/g. The first approach requires the analysis of signs measured by each test, as does the second approach. It is not clear what the difference is. Do you mean the final ‘approach’ rather than ‘stage’? The 3rd approach also needs to be explained more clearly.

86 – it is not clear why fusional vergence has been explicitly stated, whilst the others are all put under ‘signs’.

Methods

In the introduction it was stated that the students had ‘visited for primary eye care’. That should appear in the method section instead and it should be stated whether this was in a hospital setting. It is not clear how they were recruited. Was the project advertised to seek students with ocular discomfort? Or were University students identified within the primary eye care and asked to partake?

How was it determined that these students had ocular discomfort? Were they asked any questions regarding this?

96 - Were set questions asked about the ocular discomfort?

105 – replace ‘by an ocular motility and’ with ‘by ocular motility testing and’

107 - It is not clear how the phoropter was used for binocular vision tests

108 – tests were performed at distance ‘or’ near – should this be ‘and’?

112-6 – was the distance for NPC recorded in cm? How was the speed of the target controlled to 5cm/s? ‘at the eye level and between the both eyes’ should be ‘between both eyes and at eye level’.

122 – how was the speed controlled to 2PD per sec?

When measuring the accommodative amplitude, how was the speed controlled? What were the instructions? Were they told to report when the target first became blurred or when it was so blurred they could read the letters? The last word ‘accommodation’ should state ‘(in meters).

139 – this reference no. 18 seems inappropriate. Is it to justify the method being used? The use of references need to be considered as no. 17 (129) also appeared inappropriate.

142 – the word ‘prism’ is missing

143 – Relating back to my point re. 139, it would be more appropriate to have a reference here to state why testing was done at near first.

The word AC/A should always be followed by ‘ratio’. Why was 1D used? It is typically measured using 3D. Either explain or use a reference to justify the method. The distance was divided by 2.5D, presumably because the test was conducted at 40cm, but this should be stated somewhere. There was also not enough detail on how the lenses were introduced and how IPD was measured? Was this all done using the phoropter?

153 – Was the phoropter used to change lens strengths? Normative values should be presented here, and for all the other tests used.

Was BAF tested at 40cm? Normative values from the literature would also be helpful.

Vergence facility testing is the main concern of the method section. The method described does not test VF. This would require a BI and a BO prism, yet the current study used a BI and BU prism. Is this a typing error? The subject should also be reporting when the target becomes single, not when they note clear vision.

The method used to calculate Percival’s criterion is not clearly presented.

179-83 – What is Morgan’s? Is this a reference? It is not in the reference list either. So were all participants used and categorised into one of the 10 diagnoses? It is not clear what the expected criteria is for each test? Do you mean the expected normative values? These could have been presented when describing each of the tests. It is not clear what modification were made from the study by Scheiman and Wick.

Table 1 – ‘1 and 2 or 3’…does this mean you need both signs 1 and 2, or just sign 3? Or does this mean you need sign 1 as well as sign 2 or 3? If it is the first option, then perhaps ‘signs 1 and 2, or sign 3’ would be better? If it is the second option, perhaps put this as ‘presence of sign 1 and sign 2 or 3’.

Table 1 – What is meant by a ‘normal phoria’? What counts as ‘reduced’ PFV and NFV? Can abbreviate binocular accommodative facility under AI. Should BAF be more than 13 cpm in AE? Is a reference required for this table if it has been taken and modified from another source?

194-204 – Which 3 groups were compared? It has not been stated that the subjects were put into 3 groups. I had assumed subjects were put into the 10 diagnostic categories based on the information that has been provided. Should the sentence on specificity and sensitivity should come before it was stated that the ROC was calculated because the sensitivity values are used for the ROC? It would be useful to state this and 1-specificity are used to plot the curve. Perhaps I am mistaken but I think the ‘cut-off’ is the value that you use to decide what test result indicates whether or not someone has CI, and therefore would affect the sensitivity and specificity, and therefore would affect the curve. How was the AUC calculated? It would be useful to indicate that a AUC value of 1 and 0.5 are important and why.

Results

Were all the subjects with a refractive error wearing their refractive correction for testing?

211 – Here it is stated what the 3 groups are but this should have appeared in the method section. The groups are unclearly presented. What is meant by ‘all CI’? Is this anyone that fell into the CI category in Table 1? Are those in ‘BVA’ anyone that falls into any other diagnosis based on Table 1? I assume NBV are those that have nothing wrong and it might be less confusing to refer to them as the control group. The abbreviation BVA is also confusing as that typically stands for binocular visual acuity

Table 2 – The AA for All CI is reported to be 1.00D – is this correct? The significance results do not match up to those reported in the text on page 10. E.g. AA was reported to be significant on 214 but here it is not significant. NPC is reported to be significant in the table but is not reported on page 10. Phoria at distance in the table states that c is significant greater than b, but on page 10 it states that there was a significant difference between all CI and other BVA. These all need to be checked.

229 – ‘non-strabismic binocular vision anomalies’ – does this refer to everyone in all CI and other BVA but excluding those in NBV? It is confusing as the terms keep interchanging. On line 232 is the NBV group now referred to as the ‘normal group’?

232-38 Within this p/g indicate CI2AD etc so that it matches with the terms used in Table 3.

Table 3 – Again it is unclear who fits within the various groups. Does the ‘non-CI’ group contain everyone in the ‘other BVA’ group? Looking at the numbers it looks like this could be those in ‘other BVA’ and ‘NBV’. Does AD include anyone of those listed with AE and AI? I would assume so but the numbers don’t add up correctly. Why is FVD and accommodative infacility not included in this table?

246-53 – ‘for each CI test’ is unclear. Firstly, it would be better to just refer to them as ‘tests’ rather than ‘CI tests’ and secondly, they are not all included in Table 4. Again it is stated ‘for each test’ yet Figure 1 only shows the ROC curves for NPC, Sheard and Percival. State why only particular tests are presented in Table 4 and figure 1. I assume ‘for diagnostic tests excluded in Table 1’ refers to Sheard and Percival? This should be explicitly stated, rather than requiring the reader to go back and work this out.

253 – ‘AUCs were greater for NBV than for non-CI’. In an earlier comment I had stated that it seemed that ‘non-CI’ included those in ‘other BVA’ and ‘NBV’, yet now it is suggested that this is not the case.

Table 4 – How was it chosen which tests to present here? For NFV, how was it chosen whether to present break or recovery point and near or distance result? It is not clear what is meant by each of the ‘screening for various CI conditions’, e.g. are you comparing the groups, so CI3 versus NBV? Etc. The paragraph from 267 suggests you are combining the stated groups but it is not clear why you would be doing this.

Figure 1 – It is not clear which is the dotted and dashed line. Perhaps a coloured figure would be more appropriate.

301 – should state ‘was also evaluated in NBV and non-CI conditions’?

308 – the accommodative conditions were abbreviated earlier. Also, what about accommodative infacility?

311-2 – This sentence does not seem to link with the previous sentence.

327 – unclear why discriminating CI from abnormal rather than normal groups.

Reviewer #3:

This study was aimed to evaluate the ability of screening tests to discriminate convergence insufficiency, that is very important issue in the fields of heath science and related areas. The authors are presenting interesting results, however the manuscript should be improved:

1. More information about criteria to select the participants should be added in the materials method section.

2. Have research group got the informed consents from study subjects, that information should be clearly indicated in the paper

3. Why did author consider the significantly difference when A p-value of ≤ 0.05, is that should be A p-value of < 0.05?

4. The Table 1: is the criteria made by the authors, or from others? If from others these should be some citations

5. Why the figure 1 does not have the legend?

6. English errors and writing style should be checked by native English speakers

---

## [Author Response · Author response to Decision Letter 0]

10 Oct 2019

Dear reviewers:

Thank you for reviewing our manuscript PONE-D-19-17984, titled “Receiver operating characteristic curve analysis of clinical signs for screening of convergence insufficiency in young adults” submitted for publication in PLOS ONE. 

We have revised the manuscript based the reviewers’ comments. Enclosed, please find the revised manuscript with the changes highlighted in colored text.

All the authors express their appreciation for your kind consideration of our manuscript.

Sincerely,

---

## [Decision Letter · Decision Letter 1]

28 Nov 2019

PONE-D-19-17984R1

Receiver operating characteristic curve analysis of clinical signs for screening of convergence insufficiency in young adults

PLOS ONE

Dear Mr. Yu,

Thank you for submitting your manuscript to PLOS ONE. After careful consideration, we feel that it has merit but does not fully meet PLOS ONE’s publication criteria as it currently stands. Therefore, we invite you to submit a revised version of the manuscript that addresses the points raised during the review process.

We would appreciate receiving your revised manuscript by Jan 12 2020 11:59PM. To enhance the reproducibility of your results, we recommend that if applicable you deposit your laboratory protocols in protocols.io, where a protocol can be assigned its own identifier (DOI) such that it can be cited independently in the future. For instructions see: http://journals.plos.org/plosone/s/submission-guidelines#loc-laboratory-protocols

We look forward to receiving your revised manuscript.

Kind regards,

Le Hoang Son, Ph.D

Academic Editor

PLOS ONE

**Comments to the Author**

1. If the authors have adequately addressed your comments raised in a previous round of review and you feel that this manuscript is now acceptable for publication, you may indicate that here to bypass the “Comments to the Author” section, enter your conflict of interest statement in the “Confidential to Editor” section, and submit your "Accept" recommendation.

Reviewer #1: All comments have been addressed

Reviewer #2: All comments have been addressed

Reviewer #3: (No Response)

2. Is the manuscript technically sound, and do the data support the conclusions?

Reviewer #1: Partly

Reviewer #2: Yes

Reviewer #3: (No Response)

3. Has the statistical analysis been performed appropriately and rigorously? 

Reviewer #1: Yes

Reviewer #2: Yes

Reviewer #3: Yes

4. Have the authors made all data underlying the findings in their manuscript fully available?

Reviewer #1: Yes

Reviewer #2: Yes

Reviewer #3: Yes

5. Is the manuscript presented in an intelligible fashion and written in standard English?

Reviewer #1: No

Reviewer #2: Yes

Reviewer #3: Yes

6. Review Comments to the Author

Reviewer #1:

Need clarification on data distribution.

The manuscript still needs tidying up in terms of language and style (please refer to the highlighted comments in the attchment).

Reviewer #2:

Thank you for addressing my comments and feedback. The paper is much clearer. The method and results section are much easier to follow. I have no further comments.

Reviewer #3:

Thank for addressing my comments! I think you authors have worked hard to revise the manuscript as required by the reviewers

---

## [Author Response · Author response to Decision Letter 1]

12 Dec 2019

Dear Dr. Le Hoang Son:

We thank you and the reviewers for reviewing our manuscript and the valuable comments provided. We have revised our manuscript accordingly and provided the point-by-point responses to the reviewers’ comments. 

We have revised the entire manuscript based on the suggestions and the attached Reviewer #1 file.

All the authors express our appreciation for your kind consideration of our manuscript. Thank you for your consideration. I look forward to hearing from you.

Sincerely,

---

## [Decision Letter · Decision Letter 2]

14 Jan 2020

Receiver operating characteristic curve analysis of clinical signs for screening of convergence insufficiency in young adults

PONE-D-19-17984R2

Dear Dr. Yu,

We are pleased to inform you that your manuscript has been judged scientifically suitable for publication and will be formally accepted for publication once it complies with all outstanding technical requirements.

With kind regards,

Le Hoang Son, Ph.D

Academic Editor

PLOS ONE

**Comments to the Author**

1. If the authors have adequately addressed your comments raised in a previous round of review and you feel that this manuscript is now acceptable for publication, you may indicate that here to bypass the “Comments to the Author” section, enter your conflict of interest statement in the “Confidential to Editor” section, and submit your "Accept" recommendation.

Reviewer #1: All comments have been addressed

Reviewer #2: All comments have been addressed

2. Is the manuscript technically sound, and do the data support the conclusions?

Reviewer #1: Yes

Reviewer #2: (No Response)

3. Has the statistical analysis been performed appropriately and rigorously? 

Reviewer #1: Yes

Reviewer #2: (No Response)

4. Have the authors made all data underlying the findings in their manuscript fully available?

Reviewer #1: Yes

Reviewer #2: (No Response)

5. Is the manuscript presented in an intelligible fashion and written in standard English?

Reviewer #1: Yes

Reviewer #2: (No Response)

6. Review Comments to the Author

Reviewer #1: All comments have been addressed accordingly by the authors and the manuscript can be accepted for publication.

Reviewer #2: (No Response)

---

## [Editor Report · Acceptance letter]

16 Jan 2020

PONE-D-19-17984R2 

Receiver operating characteristic curve analysis of clinical signs for screening of convergence insufficiency in young adults 

Dear Dr. Yu:

I am pleased to inform you that your manuscript has been deemed suitable for publication in PLOS ONE. Congratulations! Your manuscript is now with our production department. 

With kind regards,

on behalf of

Dr. Le Hoang Son 

Academic Editor

PLOS ONE